# Whey Protein Derived Mouthdrying Found to Relate Directly to Retention Post Consumption but Not to Induced Differences in Salivary Flow Rate

**DOI:** 10.3390/foods10030587

**Published:** 2021-03-11

**Authors:** Victoria Norton, Stella Lignou, Lisa Methven

**Affiliations:** Department of Food and Nutritional Sciences, Harry Nursten Building, University of Reading, Whiteknights, Reading RG6 6DZ, UK; v.l.norton@pgr.reading.ac.uk (V.N.); s.lignou@reading.ac.uk (S.L.)

**Keywords:** mucoadhesion, whey protein, mouthdrying, whey permeate, saliva flow

## Abstract

Whey protein is fortified into beverages to provide functional benefits, however, these beverages are considered mouthdrying. To date whey protein derived mouthdrying has not been quantified using a ‘physical measure’ in parallel with rated perception. Saliva flow could also relate to whey protein derived mouthdrying, however this has not been previously tested as an intervention. Accordingly, volunteers (*n* = 40) tested mouthdrying in different whey beverages and the sensory profile was evaluated by a trained sensory panel (*n* = 10). Volunteers also rated mouthdrying combined with collection of saliva samples post beverage consumption to measure retention to the oral cavity. To modulate saliva flow rate, volunteers both chewed on parafilm (to increase saliva flow) and used cotton wool (to remove saliva) before tasting beverages and rating mouthdrying. Both the volunteers and sensory panel rated whey protein beverages (WPB) as significantly more mouthdrying than the control beverage (whey permeate). The significantly higher rating of mouthdrying from the volunteers coincided with significantly higher protein concentration in saliva samples post WPB consumption, supporting mucoadhesion as the mechanism. Modulating saliva flow did not lead to any difference in rated mouthdrying and future work would be beneficial to evaluate further the influence of natural variation in salivary flow rate.

## 1. Introduction

Protein needs are suggested to increase with age (1.0–1.2 g/kg/day) despite the current UK reference nutrient intake (RNI) for adults only being 0.75 g/kg/day [1,2]. Accordingly, there is an increasing emphasis on improving protein intake across the lifespan to offset potentially associated health conditions, slow the rate of muscle decline and promote healthy ageing [3,4]. Protein consumption is also associated with a number of positive benefits, such as improved health outcomes, appetite regulation, weight management and enhanced sports performance [1,5]. Oral nutritional supplements (ONS) and protein fortified products are often used to improve energy and protein intake especially in older adults. Whey protein is commonly fortified into these products due its associated functional benefits, such as higher leucine content and quicker digestion and absorption kinetics [6]. Products need to be an appropriate portion size, palatable, energy dense and appetising to increase successfully nutritional intake [7].

Despite the widespread recognised benefits of ONS consumption, product compliance and consumption of adequate product to meet individual needs are considered limiting factors in maximising such benefits, together with related cost and waste ramifications [7,8,9]. In addition, product palatability (for example, appearance, aroma, flavour, texture, and mouthfeel) can be a key driver of product acceptability by consumers [3]. More specifically, texture is suggested to provide a key role in food preferences, where texture awareness can relate to product expectation [10]. This is particularly relevant for dairy products as mouthfeel attributes are commonly associated with product dislike, typically build with repeated consumption and are challenging to define [9,11,12,13,14,15]. Food and beverage matrices fortified with whey protein have also resulted in negative mouthfeel attributes such as mouthdrying, hardness, slower melt rate, teeth packing, increased crumb size, chalky, mouthcoating, grainy, rough and dense [15,16,17,18]. Previously, our research group proposed whey permeate (a deproteinised whey powder) as a suitable non-protein control to fortify cakes and beverages and to provide comparisons with whey protein fortification in order to investigate mouthdrying and mucoadhesion respectively [16,19]. However, other previous studies into mouthdrying within whey protein beverage (WPB) models have typically been carried out without a non-protein control and are therefore limited in that they unable to prove if the protein within WPB is causing the mouthdrying. Understanding and addressing these proposed causes of poor compliance is key to maximising benefits from such products.

Dairy products have been associated with a ‘textural defect’ [20] often referred to as astringency, drying and mouthdrying. However, astringency is as ‘a result of exposure to substances such as alums or tannins’ [21] which are not usually present in whey protein. Accordingly, the term mouthdrying (a drying sensation in the mouth during or after consumption of a product) is considered more appropriate in the context of dairy products. The proposed causes of such whey protein derived mouthdrying remain unconfirmed and form part of our current investigation [19]. Our research group proposed mucoadhesion as a probable cause of whey protein derived mouthdrying (especially from neutral pH WPB), however further proof is required.

Mucoadhesion has been studied in drug delivery and food systems [22,23,24,25,26] and is considered in the context of this paper as the binding or sticking (retention) of whey proteins to the oral cavity [15]. Recently, our research group demonstrated that protein is considered to adhere to the oral cavity (mucoadhesion) to a greater extent post WPB consumption compared with a whey permeate beverage (WPeB), and mucoadhesion is considered to increase with age [19]. Despite establishing a valid ‘physical measure’ to measure mucoadhesion [19], a potential limitation of this previous study was that the link between mucoadhesion and mouthdrying within the same method was not investigated.

Saliva is associated with a number of key functions, such as lubrication, food clearance, taste and mouthfeel, digestion and oral health [27]. In addition, salivary flow rates are considered to reduce with age [28] and could alter sensory perception [29,30,31,32,33,34]. However, previous research into this has so far been relatively inconclusive as regards the effect on subsequent perception of protein products [16,19,35] and needs further investigation. Accordingly, understanding salivary flow changes, and its relevance to sensory perception and food acceptance, is of growing relevance.

Previous research indicates mouthdrying and mucoadhesion are present within dairy beverages and increase with consumers’ age [19,36]. However, trying to prove that the perception of mouthdrying increases with age has produced mixed results, potentially due to the lack of sensitivity of rating scales (i.e., the generalised Labelled Magnitude Scale, gLMS) compared with discrimination testing in detecting mouthdrying in older adults [19,36]. Therefore, our study will evaluate mouthdrying using both tests to explore further these concerns. The link between whether greater WPB retention results in increased WPB mouthdrying perception and the influence of salivary flow on such perception, are both relatively unclear. Accordingly, further investigation is necessary to understand these phenomena for the benefit of older adults in the future. This study hypothesises that (a) modulating salivary flow will alter mouthdrying perception and (b) oral retention is directly related to whey protein derived mouthdrying. In order to evaluate these hypotheses mouthdrying was evaluated via quantitative descriptive analysis (QDA), 2-Alternative Forced Choice Test (2-AFC) and gLMS. This study had the following objectives (a) to provide more conclusive evidence that mucoadhesion and mouthdrying of WPBs are intrinsically linked and (b) to test whether modulating saliva flow can influence perceived mouthdrying of beverages.

## 2. Materials and Methods

### 2.1. Study Overview

Forty volunteers (24.9 ± 3.4 years, healthy) completed a single blinded randomised crossover trial over two study visits (Table 1). Power calculations (alpha risk = 0.05 and 80% power) were used to calculate the subject size based on previous work in WPBs [19] using mouthdrying intensity ratings (0–100) as the primary outcome measure and, assuming a difference of 16 and standard deviation of 23, indicating the lowest sample size of 32. The study was conducted in compliance with current COVID-19 guidelines at the time (August and September 2020; with appropriate risk assessments and social distancing). The study was fully explained to the volunteers and their informed written consent was obtained prior to their participation. In addition, it was made clear that all data would be anonymised and kept confidential, as well as there being a right to withdraw. The study received a favourable opinion for conduct from the University of Reading, School of Chemistry, Food and Pharmacy Research Ethics Committee (SCFP 32/20) and the study was registered on the clinical trials database (www.clinicaltrials.gov (accessed on 11 August 2020) as NCT04507399). Volunteers were screened to ensure they met inclusion criteria (minimal medication, no COVID-19 symptoms or not having had COVID-19 within the last 4 weeks, not smokers, with no known allergies or intolerances to food, not with diabetes nor cancer and not having had oral surgery or a stroke). The study visits (Figure 1) were held at the Sensory Science Centre, University of Reading.

### 2.2. Materials

Two whey powders were used: whey protein concentrate (WPC) (Volactive Ultra-Whey 80; minimum protein content 80%, remainder as lactose, fat, moisture and ash) and whey permeate (WPe) (Volactose Taw Whey Permeate; minimum lactose content 89%, remainder ash, moisture, protein and fat) (Volac International Ltd., Royston, UK). Sucrose (Caster sugar, Tate & Lyle, London, UK) and vanilla extract (Nielsen-Massey, The Netherlands) were sourced from Sainsbury’s (Reading, UK). Parafilm^®^, Bradford reagent (0.1–1.4 mg/mL) and protein standard (Bovine Serum Albumin, BSA, 2 mg/mL) were purchased from Sigma-Aldrich (Dorset, UK).

### 2.3. Model Beverage Preparation

Four different whey beverages were tested: (1) a control whey permeate beverage (WPeB; 4% *w*/*v*, WPe powder in deionised water) and (2) a whey protein beverage (WPB; 10% *w*/*v*, WPC powder in deionised water). The rationale was as outlined in our previous work [19]. WPe provides a non-protein whey control at a concentration selected to keep the lactose content below sweet taste recognition and WPC concentration is relevant to commercial products as well as commonly utilised in WPB testing [15,37,38,39]. In addition, sample palatability was improved by adding sucrose and vanilla as previous work highlighted that unsweetened WPBs were rated as disliked moderately (mean 3 on 9-point hedonic scale) [19]. This resulted in (3) a sweetened control whey permeate beverage (WPeBS; 4% *w/v* WPe, 1.49% *w/v* sucrose, 1% *w/v* vanilla extract) and (4) a sweetened whey protein beverage (WPBS; 10% *w/v* WPC powder, 2% *w/v* sucrose, 1% *w/v* vanilla extract). Less sucrose was added to the WPeBS compared with the WPBS due to the lactose content of the WPe; they were matched on relative sweetness. Formulations are summarised in Table 2. Samples were prepared simultaneously and stirred (magnetic stirrers at medium speed; Stuart^TM^ SM5, Cole-Parmer, Staffordshire, UK) for 90-min at room temperature (21.8 ± 2.0 °C). Samples were left to hydrate overnight at 4 °C before being served to volunteers at room temperature.

### 2.4. Sensory Methods

All sensory evaluation (trained sensory panel and volunteers) was carried out under red lights (to mask minor visual differences between samples) in isolated booths using Compusense Cloud Software (Version 21.0.7713.26683, Compusense, ON, Canada). Palate cleansing between samples used filtered warm water [36]. All samples were presented at the same time on different trays (due to COVID-19 serving restrictions) but tasted in a randomly allocated sequential balanced order, and coded with a random three-digit number. Samples (10 mL) were presented in black plastic cups (25 mL; opaque) (BB Plastics, West Yorkshire, UK).

### 2.5. Sensory Profile

Quantitative descriptive analysis (QDA™) [40] was used to determine the sensory differences between the whey beverages, as well as to quantify the attribute changes arising from the addition of sucrose and vanilla. All panellists (*n* = 10; 9 female and 1 male, screened and trained) had a minimum of one years’ experience and at least six hours training involving whey beverages. Both trained panel and study volunteers had the same samples (WPeB, WPeBS, WPB and WPBS). The trained panel developed a consensus vocabulary (adapted from previous work [15]) identifying 21 attributes as outlined in Table 3. Appearance was not evaluated due to potential visual differences between samples which could lead to bias evaluation; accordingly to address these concerns samples were presented in opaque black plastic cups and under red lights to minimise such differences between samples. Panellists evaluated the samples in duplicate (in different sessions) using unstructured line scales (0–100) with appropriate anchors.

### 2.6. Mouthdrying 2-Alterative Forced Choice Test (2-AFC)

Volunteers were provided with clear instructions, presented with two samples and asked which sample was more mouthdrying via a single paired comparison test comparing WPeBS with WPBS (in accordance with ISO 5495:2005) [41]. The rationale for using a 2-AFC test was due to its simplicity and ability to detect small differences between samples; it had previously been used successfully to find such differences between products [36,42].

### 2.7. Whey Beverage Individual Perception and Liking

Volunteers rated liking (9-point hedonic scale), easiness to drink and swallow (5-point category scale), attribute perception (logarithmic scale (gLMS) with descriptors for intensity of sweetness, thickness and mouthdrying attributes), appropriateness of attribute level (Just-About-Right, JAR; 5-point hedonic JAR scale), preference and consumption of whey beverages (ranked preference; a series of 2-AFC tests to assess paired preference; frequency of consumption on 6-point category scale) and provided comments relating to flavour and texture. All volunteers completed a training exercise (Appendix A; rating 15 remembered or imagined sensations adapted from previous work [43]) to become familiar with gLMS [44] as a scale. Volunteers had a break (45 s) between samples during which they cleansed their palate by drinking warm filtered water.

### 2.8. Modulating Saliva Flow and Mouthdrying Perception

To understand the role of saliva on mouthdrying perception (Figure 2); saliva flow was modulated for 2-min by either decreasing saliva flow via placing 4 × cotton wool rolls (40 mm × 10 mm) (two on each side split between the upper and lower jar) within the mouth or increasing saliva flow by chewing on parafilm^®^ (5 × 5 cm) (adapted from previous work [31,45]). Volunteers were given four 10 mL beverage samples (2 × WPeBS and 2 × WPBS) and immediately following consumption scored the sample for mouthdrying on a gLMS as well as scoring the aftereffects of mouthdrying at 15 s, 30 s, 60 s and 120 s time intervals post consumption. Volunteers also had an enforced 3-min break between samples (rationale based on initial testing within our lab and protein concentration in saliva samples post WPB consumption being considered to have plateaued within 3-min) [15,19], where they swilled and consumed warm filtered water.

### 2.9. Salivary Flow Rates

Unstimulated and stimulated saliva were both collected at the beginning of study visit two with a sufficient rest (~10 min) in between. Saliva collection methods were as outlined in our previous work [16,19]. In summary, unstimulated saliva was collected for 5-min whereas stimulated saliva was collected for 2-min whilst chewing parafilm^®^ (5 × 5 cm). Saliva samples were collected in tubes (60 mL, wide) and flow rates calculated as mL/min. Samples were stored on ice before analysis.

### 2.10. Saliva Samples Post Beverage Consumption and Mouthdrying Perception

An oral retention method from Norton et al. [19] was developed to measure protein retained in saliva after swallowing, alongside rating of mouthdrying (Figure 3). Stimulated saliva samples were collected (as outlined in Section 2.8) and used as a baseline measurement (rationale based on previous work [19]). Eight beverage samples (4 × WPeBS and 4 × WPBS; 10 mL) were provided at two time points (15 s and 60 s, randomised). These were considered key time points based on previous work [15,19]). Volunteers (on eight occasions) gave four saliva samples and rated four beverages for perceived mouthdrying on a gLMS post beverage consumption. A 5-min break was obligatory between samples to prevent crossover effects and ensure protein concentration in saliva samples had plateaued [19,46]. Warm filtered water was consumed to palate cleanse during this break. Tubes were weighed before and after collection to measure saliva weight and all saliva samples were stored on ice pending analysis.

### 2.11. Protein Analysis of Saliva Samples

Bradford Assay was used to analyse the protein concentration (mg/mL) in saliva samples [47,48] as described in Norton et al. [19]. In summary, all analysis was performed in triplicate with biological and analytical replicates. BSA was used as the protein standard (6 dilutions; 0.125 to 2 mg/mL). Saliva samples diluted 1:2 (saliva: purified water) and analysis followed immediately after each volunteer’s visit. Volunteers baseline values (i.e., protein concentration in stimulated saliva) were subtracted from sample measurements to calculate protein concentration remaining post WPBS consumption. WPeBS was used as a control beverage and as outlined in previous work [19]; the protein concentration was already below the baseline value (i.e., stimulated saliva protein concentration) therefore no additional calculations were required.

### 2.12. Statistical Analysis

Analysis of variance (ANOVA) was used to analyse sensory profile data [49,50] with main effects tested against the sample by assessor interaction, sample fitted as a fixed effect and assessor as a random effect using SenPAQ software (version 5.01, Qi Statistics, Kent, UK). Fisher’s least significant difference (LSD) was used to test sample pairs assuming a 5% significance level.

Mouthdrying 2-AFC data was analysed using Binomial expansion and Thurstonian modelling in V-power to calculate *p* values, power and d’ value [51]. Quantile analysis (based on the median) grouped volunteers into low and high salivary flow rates (XLSTAT version 2020.1.3, Addinsoft, Paris, France). Perception and liking data from volunteers were analysed via linear mixed models using explanatory variables of sample, sex, saliva flow, volunteers fitted as a random effect, with the dependent variables of attribute perception, liking and JAR rating scores (SAS^®^ software, version 9.4, Cary, NC, USA, applying Bonferroni). Volunteers modulated saliva flow and mouthdrying perception data were analysed with sample, time, condition, sex, and saliva flow as explanatory variables, volunteers fitted as a random effect, and mouthdrying perception rating as the dependent variable. Salivary flow rates and baseline saliva samples were analysed using explanatory variable of sex, volunteers fitted as a random effect and with the dependent variables of saliva flow and protein concentration respectively. Volunteers saliva samples post beverage consumption and mouthdrying perception data were analysed with the explanatory variables of sample, time, sex, saliva flow, volunteers fitted as random effects and with protein concentration and mouthdrying perception as the dependent variables. All attribute data was collected on the gLMS log-scale and was transformed to linear data (anti-logged). Data reflects least square means (LSM) estimates.

Penalty analysis of the JAR and liking data was carried out (as previously described [16,19]) using XLSTAT. Paired preferences were analysed using Binomial expansion in V-Power [51]. It should be noted that only two volunteers were taking medication and therefore outlier analysis was conducted using a Dixon test in XLSTAT. Outlier analysis demonstrated that these volunteers were not considered outliers (except for one volunteer for one output measure (thickness)). Analysis was therefore carried out with and without this volunteer’s data, with the overall result being the same and accordingly all data was included within the statistical analysis. Significant differences were defined in all analyses by *p* < 0.05.

## 3. Results

### 3.1. Sensory Profile

The sensory profile demonstrated that 12 of the 21 attributes were significantly different (*p* < 0.05) between samples as outlined in Table 3. In summary, it demonstrated whey protein beverages (WPB and WPBS) significantly increased mouthdrying, chalky and body compared with whey permeate beverages (WPeB and WPeBS). Adding sucrose and vanilla to beverages (WPeBS and WPBS) resulted in significantly increased sweet and vanilla notes compared with WPeB and WPB, as well as significantly reduced mouthdrying in WPBS compared with WPB, therefore improving sample palatability.

### 3.2. Mouthdrying 2-Alterative Forced Choice Test (2-AFC)

The mouthdrying paired comparison test demonstrated that WPBS was significantly more mouthdrying (*p* < 0.0001; d’ value: 1.19; power: 0.99) compared with WPeBS; 60% of the volunteers were able to distinguish that WPBS was more mouthdrying.

### 3.3. Whey Beverage Individual Perception and Liking

Volunteers perceived WPBS as significantly (*p* < 0.05) more mouthdrying, thicker, less sweet and less easy to consume compared with WPeBS (Figure 4 and Figure 5). There was no significant difference (*p* = 0.53) in liking between whey beverages with both beverages perceived, on average, as neither like nor dislike on a 9-point hedonic scale. There was also no significant difference in Just-About-Right flavour and thickness between whey beverages, where both were perceived as closer to Just-About-Right (JAR = 3) compared with too weak/thin for flavour and thickness respectively (Table 4). Saliva flow had no significant effect on whey beverage liking, perception, easiness to consume or JAR attributes, whether it was tested as overall or by grouping volunteers into low and high saliva flow (Appendix A). There was also no significant effect of sex on whey beverage individual perception and liking (Appendix A).

There was no significant difference (*p* = 0.13) in preference between whey beverages. However, this study successfully demonstrated improvements in sample palatability compared with previous samples, as WPeBS and WPBS were both significantly preferred (*p* < 0.0001) compared with WPeB and WPBS (Table 5). Where attributes are not at the optimum level for a volunteer (as reflected in Just-About-Right, JAR, ratings) this may impact liking. The penalty analysis (Table 4) concluded liking was negatively impacted where flavour was considered too low. Volunteers generally provided positive feedback for flavour and texture of both beverages; 86 comments were provided of which 53 were positive and 33 were negative (Table 6).

### 3.4. Modulating Saliva Flow and Mouthdrying Perception

Modulating saliva flow led to no significant change (*p* = 0.96) in perceived mouthdrying, as mouthdrying perception remained relatively consistent within each beverage type (Figure 6). In common with the results where saliva was not modulated, there was a significant effect of sample (*p* < 0.0001) where WPBS was more mouthdrying compared with WPeBS at all timepoints (0 s, 15 s, 30 s, 60 s and 120 s) (Figure 6). Time also had an overall significant effect (*p* = 0.0002) where perceived mouthdrying slightly increased over time (Figure 6). There was no significant effect of saliva flow and sex on mouthdrying perception following modulated saliva flow (Appendix A).

### 3.5. Salivary Flow Rates

Unstimulated salivary flow rates was 0.72 ± 0.04 mL/min, whereas stimulated flow was 2.29 ± 0.11 mL/min. Volunteers were also categorised by quantile analysis into low and high salivary flow rates (Table 1). There was no significant effect of sex (unstimulated saliva flow (USF): *p* = 0.15 and stimulated saliva flow (SSF): *p* = 0.053) on saliva flow regardless of collection method. However, there was a tendency for males to have a higher salivary flow compared with females (USF: males 0.81 ± 0.09 and females 0.68 ± 0.05 mL/min and SSF: males 2.61 ± 0.20 and females 2.15 ± 0.13 mL/min).

### 3.6. Saliva Samples Post Beverage Consumption and Mouthdrying Perception

WPBS led to a significantly higher protein concentration (*p* < 0.001) in saliva samples post swallow compared with WPeBS at both timepoints (15 s and 60 s) (Figure 7). There was no significant effect of time overall on protein concentration in saliva samples post beverage consumption (*p* = 0.052), however, there was a significant time by sample interaction (*p* = 0.03). Pairwise comparison highlighted that WPBS consumption resulted in saliva samples showing a significantly higher (*p* = 0.003) protein content at 15 s compared with 60 s, whereas WPeBS had a lower saliva protein content across all timepoints (*p* = 0.83) (Figure 7). Results from the saliva samples post beverage consumption supported the mouthdrying perception results, where WPBS resulted in significantly higher mouthdrying scores (*p* < 0.001) compared with WPeBS at both timepoints (Figure 7). However, there was no overall significant effect of time (*p* = 0.26) on perceived mouthdrying where WPBS decreased very slightly over time whereas WPeBS remained relatively consistent (Figure 7). There were no significant effects of protein concentration in saliva samples and mouthdrying perception relating to sex or saliva flow (Appendix A).

## 4. Discussion

### 4.1. Sensory Profile and Whey Beverage Individual Perception and Liking

Fortifying beverages with whey protein increased mouthdrying, chalky, thickness, body and reduced sweetness and easiness to consume compared with a non-protein control (in this case a whey permeate beverage (WPeB)). These findings support previous work in this area that WPBs are associated with mouthdrying, mouthcoating and chalky attributes [14,15]. These studies were however carried out without a non-protein control, therefore our study concluded that it is indeed the protein in WPBs, rather than other constituents of whey, that cause mouthdrying within WPBs. Previous research highlighted the lack of sensitivity of a gLMS (0–100) compared with a 2-AFC in detecting mouthdrying in older adults [19,36]. Accordingly, to address these concerns, our study also measured mouthdrying using a paired comparison test to a ensure differences between samples were not missed on a gLMS (0–100), which can occur if samples are presented monadically [52]. The 2-AFC clearly demonstrated the majority of the volunteers (32 out of 40) supported WPBS as being more mouthdrying compared with WPeBS. Therefore, our study proved volunteers perceived WPBS as more mouthdrying compared with WPeBS (by both gLMS and 2-AFC), which was additionally supported by the trained panel findings. A limitation of our study was not being able to recruit older adults due to the ongoing COVID-19 pandemic. Accordingly, next steps should include future work with older adults to prove conclusively that sensitivity to mouthdrying increases with age, using a more sensitive discrimination test (i.e., 2-AFC) in different food matrices.

Previous work by Norton et al. [19] demonstrated low liking scores by volunteers for model WPBs, therefore this study added sucrose and vanilla to improve potentially flavour and acceptability. The sensory profile concluded that adding sucrose and vanilla increased sweet and vanilla notes, which subsequently reduced perception of mouthdrying. This did lead to an improvement in volunteers’ liking ratings and a clear preference for the ‘improved beverages’. The addition of sucrose and flavour led to increased product acceptance and reduced perceived mouthdrying such additions are commonly found in commercial oral nutritional supplements (ONS). However, the sweetened WPB (WPBS) was still mouthdrying and further mitigation may lead to increased palatability. This could maximise product benefits, especially as these products are most often consumed by older adults who may be more sensitive to the products oral adhesion [19] and mouthdrying [36].

It was hypothesised that WPBs would cause mouthdrying, thereby reducing beverage acceptability. However, surprisingly, there was no difference in liking or preference between the two beverages (WPeBS and WPBS). This could be explained by the WPeBS where volunteers lack familiarity with the product and highlighted its minimal flavour, watery, thin and sweet nature, as demonstrated by volunteers consumption habits, penalty analysis and comments. These findings were also supported by the trained panel who identified sweet and vanilla taste, as well as being lower in cooked notes (such as cooked milk and butter) in the WPeBs compared with WPBs. Furthermore, sweetness and thickness are considered key drivers of acceptability in milk beverages [53], which could explain the relatively low liking scores and no difference in liking or preference between the beverages demonstrated in our study.

The sensory profile demonstrated that the WPBS had more body compared with the WPeBS and this result was matched by the volunteers who also perceived WPBS to be thicker. Although the viscosity of the beverages was considered to be broadly similar (Appendix A), there was a mean difference of 0.83 mPaS at 50 s^−1^ (a commonly cited oral shear rate) [54]. A previous study has shown the Weber fraction (K) for oral thickness perception of model beverages to be 0.26 [55], and therefore with the WPBS thickness at 1.78 mPas, the calculated just noticeable difference (JND) would be 0.46 mPas. Hence the literature supports that there would be a perceptual difference in thickness between the WPeBS and WPBS. However, a previous study measuring astringency of low pH WPBs used maltodextrin to modify viscosity (1.6 to 7.7 mPAS) and found it had no effect on perceived astringency [56]. This supports our current study in that the noticeable difference in thickness is unlikely to have influenced perception of mouthdrying, however the previous study utilised a low pH whey model, where it is likely that the mechanism of astringency was different to the mechanism of mouthdrying proposed in our neutral pH samples (mucoadhesion) [15,19]. Therefore, it is advisable that future work aims to ensure viscosity is fully matched between beverages (potentially by using hydrocolloids). However, it may be challenging to match such low viscosities and in addition the use of hydrocolloids may potentially alter taste, flavour and mouthfeel properties [26] and lead to a different viscosity response to shear, compared with the viscosity profile resulting from protein.

### 4.2. Modulating Saliva Flow and Mouthdrying Perception

There are numerous key functions associated with saliva [27] and saliva can influence sensory perception. Therefore, it was hypothesised that modulating salivary flow by either decreasing or increasing saliva flow would alter mouthdrying perception. However, no changes in mouthdrying perception were demonstrated immediately post beverage consumption nor over time (as evidenced from the aftereffects) as a result of modulating saliva flow. These findings support previous work which has demonstrated no, or only a minimal, effect of saliva flow on perception of other sensory attributes. For example, modifying salivary flow rates (unstimulated saliva flow and stimulated saliva flow using odour, parafilm and citric acid) had no effect on sensory ratings (8 attributes: flavour (vanilla, bitter/chemical), mouthfeel (temperature, thickness, melting, creaminess) and afterfeel (fat, astringent) of custard desserts [57]. In addition, artificially increasing saliva (by adding saliva related fluids to the product) had minimal effect on sensory perception (apart from increasing melting and decreasing thickness, creaminess and fatty afterfeel sensations) of custard desserts [30]. Therefore, neither different salivary flow rates nor artificially increasing saliva volume had previously resulted in substantial differences in sensory perception in semi solid foods. Salivary composition (total protein concentration and amylase activity) has been shown to alter texture perception of custard desserts & mayonnaise [58]. However, more recently, Crawford and Running [59] demonstrated changes in salivary proteins (proline-rich proteins and cystatins) had only minimal effects on the sensory perception of chocolate milks. Vandenberghe-Descamps et al. [29] also demonstrated very few effects from differences in saliva flow on perception; they proposed that individuals may adapt their food oral processing to compensate for differences in saliva flow status and which may result in little impact on subsequent perception.

Within plant derived food models (such as tea and wine), saliva is considered to influence astringency perception. For example, volunteers with low salivary flow rates perceived wines to be more astringent over a longer duration compared with those with higher salivary flow rates [32]. Whereas after consuming black tea, perceived astringency has been shown to increase with decreasing saliva flow (by washing with water) and decrease with increasing saliva flow (by chewing on parafilm) [31]. However, these findings were not demonstrated in our study using whey beverages; this is likely to be as a result of the different mechanism involved in astringency (i.e., polyphenols binding to salivary proteins) compared with mouthdrying in neutral pH beverage (i.e., oral retention or mucoadhesion) and accordingly mechanisms may respond differently to salivary flow rate.

In addition, our study decreased saliva flow by using cotton wool rolls within the mouth (rather than washing with water) to replicate the ‘dry’ feeling within the mouth (a method successfully utilised previously by Brunstrom et al. [45]). Such findings could suggest the role of saliva flow on sensory perception is potentially food model specific and dependent on the underlying mechanism responsible for the mouthdrying sensation. Accordingly, further research is necessary to understand the role of saliva flow on mouthdrying perception in whey protein food models as current research has resulted in minimal differences so far. This could relate to how studies have measured or modulated saliva flow and is therefore potentially not a true reflection of natural variation in saliva.

### 4.3. Saliva Samples Post Beverage Consumption and Mouthdrying Perception

Whey protein adhered to the oral cavity (oral retention as a marker of mucoadhesion) post WPBS to a greater extent compared with WPeBS, supporting previous work in this area [19]. Furthermore, our study demonstrated that perceived mouthdrying was significantly increased following WPBS consumption compared with WPeBS, which matched the oral retention results. Retention declined over time, however, this trend was not matched by perceived mouthdrying which did not reduce significantly over time. Previously, a build-up of whey protein derived mouthdrying was suggested to be as a result of a possible mucoadhesion mechanism [15,19,46,60]. Mucoadhesion within a WPB is considered to be as a result of the following potential mechanisms [61]:(1)movement of the sample in the mouth provides a greater surface area for whey protein to adhere to the oral cavity;(2)spreading and swelling on the oral mucosa leads to increased adhesion and stronger adhesive joint via different physiochemical interactions [22,25];(3)mucoadhesion is considered to result from a prolonged oral exposure and loss of saliva lubrication and increased friction, tissue exposure, adhesion and interaction [46,62] can result in perceived mouthdrying potentially caused by mucoadhesion.

Therefore, our study reinforces the suggestion that mucoadhesion could be a cause of whey protein derived mouthdrying, as this study measured for the first time both oral retention of protein and mouthdrying within the same protocol and demonstrated both increased retention and perceived mouthdrying following WPB consumption. This study aimed to quantify mouthdrying using a ‘physical measure’ (i.e., retention as a measure of mucoadhesion) at the same time as scoring mouthdrying perception within WPBs, as no previous study to our knowledge has investigated this. Typically, correlations are found in the literature between potential mechanisms and sensory data and this can result in an inability to prove relationships which should be a key priority for ongoing research. Future work however remains necessary to prove mucoadhesion is the cause of the oral retention and to demonstrate that a reduction in retention would lead to a subsequent decrease in perceived mouthdrying.

## 5. Conclusions

This study demonstrated by using three different methods (QDA, 2-AFC and gLMS) that WPBs were significantly more mouthdrying compared with WPeBs. In addition, increasing sweetness in WPBs significantly reduced perceived mouthdrying and increased consumer preference. Such results suggest improving mouthfeel attributes associated with WPBs could be a key strategy to improve compliance and product suitability for older adults. This study was unable to demonstrate a role of saliva flow on mouthdrying perception. However further research using improved methodology that captures the natural variation in saliva flow is needed to understand better the impact of salivary flow changes on mouthdrying perception in whey protein food models. Previously, mucoadhesion had been considered as a probable cause of whey protein derived mouthdrying and our study highlighted WPB consumption significantly increased oral retention of the protein, which coincided with perceived mouthdrying. Hence, we conclude that whey protein is the cause of WPB retention and mouthdrying. Mucoadhesion is the probable cause of whey protein derived mouthdrying and oral retention provides a physical measure of perceived mouthdrying. However, it still needs further proof that modulating retention would result in changes in perceived mouthdrying. Understanding such mechanisms could result in improved products and increased consumption, this is important as protein consumption is associated with numerous benefits. There is a growing emphasis on improving protein intake across the lifespan to enhance health outcomes and given the potential importance of WPBs in achieving this, they must have high palatability to promote consumption and maximise the benefits from protein products.

## Figures and Tables

**Figure 1 foods-10-00587-f001:**
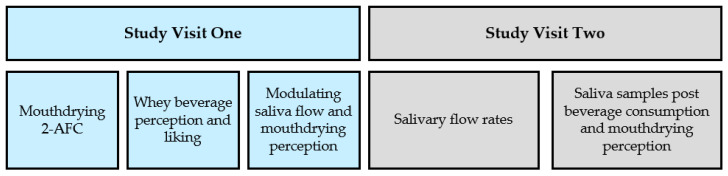
Study Overview (2-AFC: two alternative forced choice).

**Figure 2 foods-10-00587-f002:**
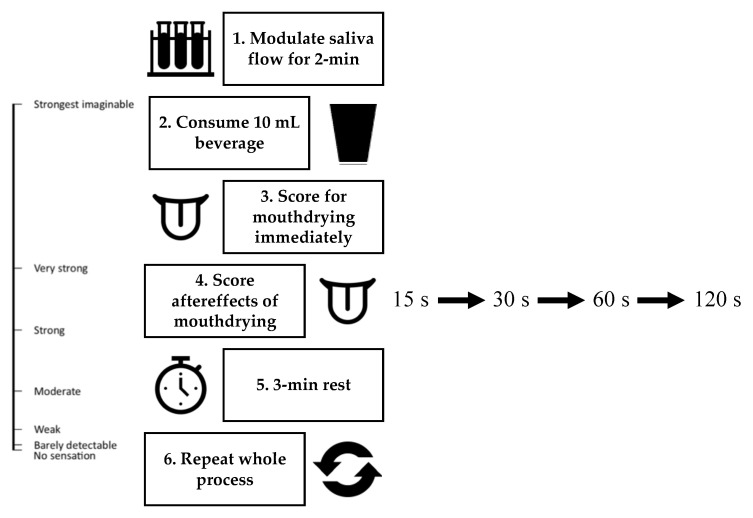
Brief overview of modulating saliva flow and mouthdrying perception protocol.

**Figure 3 foods-10-00587-f003:**
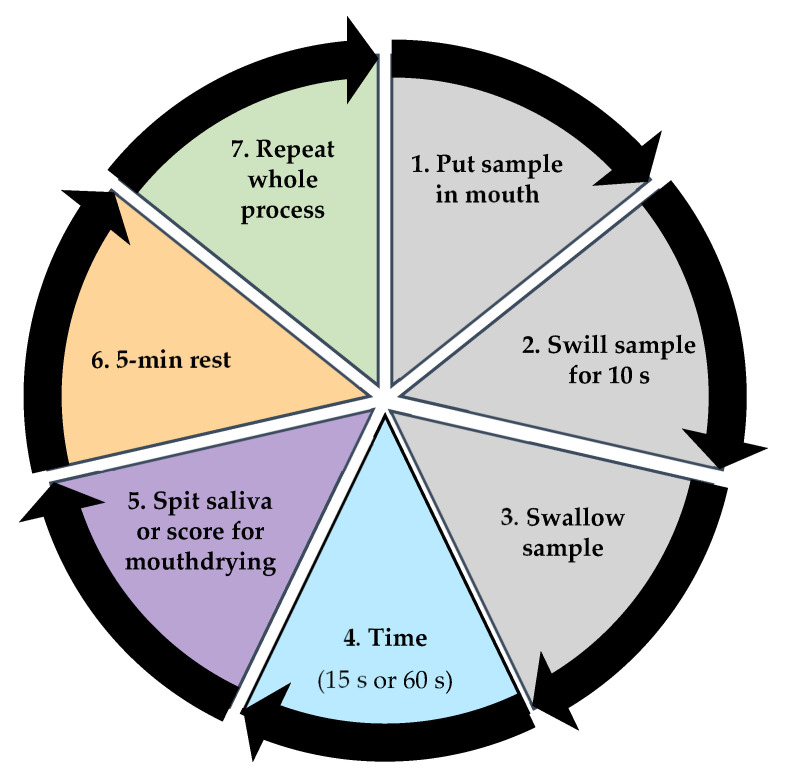
Summary of protocol for saliva sample collection and mouthdrying perception rating post beverage consumption.

**Figure 4 foods-10-00587-f004:**
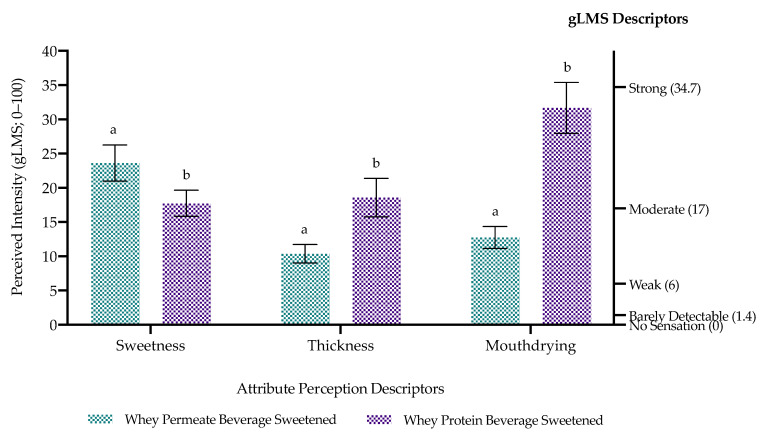
Volunteers’ attribute perception mean ratings (±standard error) of whey beverages (*n* = 40; anti-logged data). Sample significant differences denoted by differing small letters.

**Figure 5 foods-10-00587-f005:**
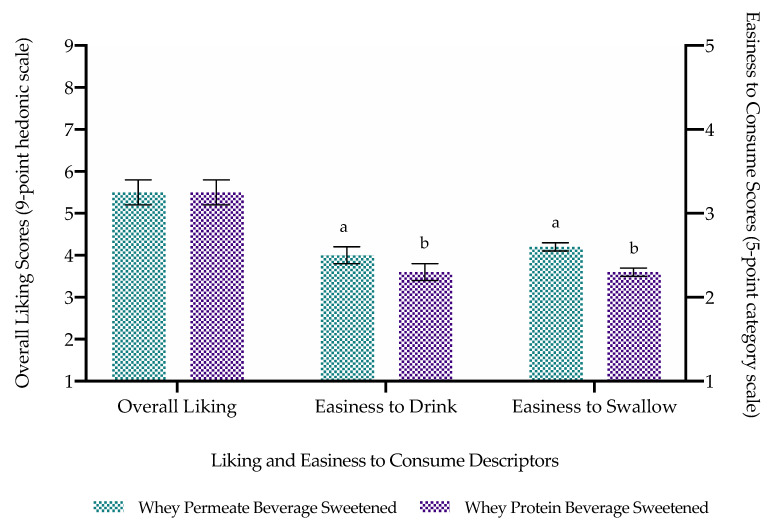
Volunteers’ liking (left axis; measured on a 9-point hedonic scale) and easiness to consume (drink or swallow) (right axis; measured on a 5-point category scale) mean ratings (±standard error) of whey beverages (*n* = 40). Sample significant differences denoted by differing small letters.

**Figure 6 foods-10-00587-f006:**
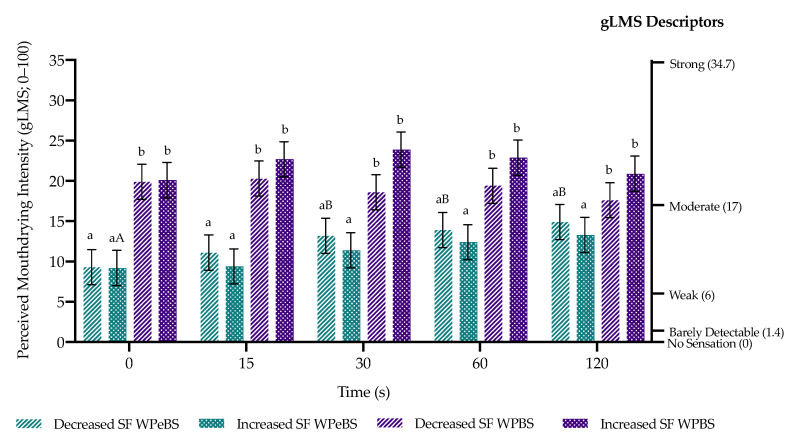
Volunteers’ perceived mouthdrying (±standard error) post beverage (WPeBS: whey permeate beverage sweetened; WPBS: whey protein beverage sweetened) consumption over time following saliva flow (SF) being modulated (increased: chewing on parafilm and decreased: by placing cotton wool rolls within the mouth). Sample significant differences are represented by differing small letters (between samples) and capital letters (within samples).

**Figure 7 foods-10-00587-f007:**
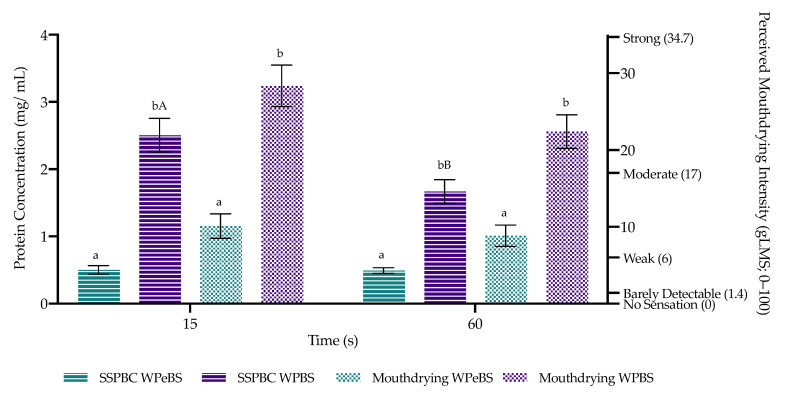
Protein concentration in saliva samples post beverage consumption (SSPBC) (left axis) and perceived mouthdrying (right axis; measured on gLMS (0–100)) (±standard error). WPeBS: whey permeate beverage sweetened and WPBS: whey protein beverage sweetened. Sample significant differences are represented by differing small letters (between samples) and capital letters (within samples).

**Table 1 foods-10-00587-t001:** Summary of volunteers’ sex, medication and salivary flow rates categories (‘*n*’ and ‘%’ indicate number and percentage). Saliva flow categories were defined as below (or equal to) or above the median (missing data *n* = 1).

Variable	Total (*n* = 40)	
	*n*	%
**Sex**		
Male	12	30
Female	28	70
**Medication**		
Yes	2	5
No	38	95
**Unstimulated Saliva Flow (mL/min)**		
Low (0.10–0.70)	19	49
High (0.70–1.4)	20	51
**Stimulated Saliva Flow (mL/min)**		
Low (0.78–2.23)	21	54
High (2.23–4.08)	18	46

**Table 2 foods-10-00587-t002:** Composition of whey beverages (WPeB: whey permeate beverage; WPeBS: whey permeate beverage sweetened; WPB: whey protein beverage; WPBS: whey protein beverage sweetened) per 10 mL (as tasted) and per 100 mL.

	WPeB		WPeBS		WPB		WPBS	
	Per 10 mL	Per 100 mL	Per 10 mL	Per 100 mL	Per 10 mL	Per 100 mL	Per 10 mL	Per 100 mL
Energy (kcal)	1.5	14.7	2.4	23.7	4.0	39.7	5.1	50.7
Fat (g)	0.0008	0.008	0.0008	0.008	0.07	0.7	0.07	0.7
of which saturates (g)	-	-	-	-	0.03	0.3	0.03	0.3
Carbohydrate (g)	0.4	3.6	0.5	5.1	0.04	0.4	0.2	2.4
of which sugars (g)	0.4	3.6	0.5	5.1	0.04	0.4	0.2	2.4
Protein (g)	0.01	0.1	0.01	0.1	0.8	8.2	0.8	8.2
Moisture (g)	0.004	0.04	0.004	0.04	0.05	0.5	0.05	0.5
Ash (g)	0.02	0.2	0.02	0.2	0.04	0.4	0.04	0.4

Composition was calculated from technical data sheets of ingredients used. The viscosity of beverages was measured and considered broadly similar, as outlined in Appendix A.

**Table 3 foods-10-00587-t003:** Sensory profile (means of two replicates ± standard error) of whey beverages (WPeB: whey permeate beverage; WPeBS: whey permeate beverage sweetened; WPB: whey protein beverage; WPBS: whey protein beverage sweetened).

Modality	Attribute	Reference and/or Description	WPeB	WPeBS	WPB	WPBS	Significant of Sample (*p* Value)
Aroma	Cooked milk	Heated pasteurised semi-skimmed milk	9.2 ± 2.7	8.1 ± 2.9	20.6 ± 4.4	18.4 ± 3.9	0.12
	Powdered milk (wet)	Skimmed milk powder (10% *w/v*, skimmed milk powder in deionised water)	7.7 ± 2.6	20.7 ± 3.9	11.9 ± 3.9	17.8 ± 3.8	0.07
	Whey isolate	Volactive Ultra-Whey 90 Instant (5% *w/v*, WPI powder in deionised water)	8.8 ± 2.4	6.3 ± 3.8	7.6 ± 2.8	10.1 ± 3.7	0.80
	Vanilla	Vanilla extract (Nielsen-Massey)	0.7 ± 1.9 ^c^	42.1± 5.1 ^a^	1.1 ± 1.9 ^c^	31.8 ± 4.8 ^b^	<0.0001
Flavour	Sour	Citric acid (0.76 g/L)	17.5 ± 3.5 ^a,b^	8.0± 4.9 ^b^	23.9 ± 4.0 ^a^	17.5 ± 4.9 ^a,b^	0.048
	Metallic	Iron (II) sulphate heptahydrate (0.0036 g/L)	8.7 ± 3.3	8.2 ± 2.5	10.1 ± 3.7	5.9 ± 3.7	0.44
	Salty	Sodium chloride (1.19 g/L)	7.7 ± 2.2	5.0 ± 2.2	9.4 ± 2.6	6.3 ± 1.9	0.27
	Sweet	Sucrose (5.76 g/L)	19.6 ± 3.0 ^b^	52.2 ± 6.4 ^a^	12.1 ± 2.5 ^b^	46.6 ± 5.8 ^a^	<0.0001
	Cooked butter	Melted unsalted butter	9.6 ± 3.0	3.3 ± 6.6	9.8 ± 2.6	9.7 ± 6.0	0.43
	Cooked milk	Heated pasteurised semi-skimmed milk	15.2 ± 3.3	12.1 ± 2.9	24.4 ± 4.0	24.3 ± 4.4	0.17
	Powdered milk (wet)	Skimmed milk powder (10% *w*/*v*, skimmed milk powder in deionised water)	6.1 ± 3.4	16.4 ± 3.8	14.3 ± 4.3	19.2 ± 4.1	0.12
	Whey isolate	Volactive Ultra-Whey 90 Instant (5% *w*/*v*, WPI powder in deionised water)	14.7 ± 2.8	8.6 ± 3.8	17.5 ± 3.4	14.2 ± 4.1	0.32
	Vanilla	Vanilla extract (Nielsen-Massey)	2.5 ± 2.6 ^b^	41.3 ± 5.3 ^a^	0.0 ± 2.9 ^b^	33.5 ± 5.0 ^a^	<0.0001
Mouthfeel	Body	Fullness of sample	21.0 ± 3.3 ^b^	21.4 ± 4.2 ^b^	31.2 ± 4.6 ^a^	31.4 ± 4.2 ^a^	0.006
	Chalky	Dry fine insoluble powder	4.3 ± 3.4 ^b^	3.9 ± 3.1 ^b^	27.3 ± 5.1 ^a^	16.8 ± 3.8 ^a^	0.0003
	Mouthdrying	Drying sensation in the mouth	26.5 ± 4.1 ^c^	30.3 ± 4.5 ^c^	51.2 ± 6.3 ^a^	42.7 ± 4.5 ^b^	<0.0001
Aftertaste	Aftertaste strength	The strength of the overall aftertaste	17.9 ± 3.3 ^b^	38.1 ± 4.0 ^a^	23.7 ± 5.1 ^b^	38.2 ± 3.6 ^a^	<0.0001
	Mouthdrying	Drying sensation in the mouth	24.6 ± 2.8 ^b^	30.2 ± 4.3 ^b^	50.4 ± 4.6 ^a^	44.0 ± 3.6 ^a^	<0.0001
	Metallic	Iron (II) sulphate heptahydrate (0.0036 g/L)	4.9 ± 3.3 ^b^	3.3 ± 4.7 ^b^	9.2 ± 5.8 ^a^	5.7 ± 5.2 ^a,b^	0.02
	Vanilla	Vanilla extract (Nielsen-Massey)	1.7 ± 1.1 ^b^	27.4 ± 4.1 ^a^	0.0 ± 1.8 ^b^	26.7 ± 4.8 ^a^	<0.0001
	Sweet	Sucrose (5.76 g/L)	12.7 ± 2.2 ^b^	35.6 ± 3.8 ^a^	7.5 ± 1.9 ^b^	34.2 ± 5.0 ^a^	<0.0001

The trained panel (*n* = 10) scored all samples in duplicate in separate sessions and data was collected using unstructured line scales (0–100). Sample significant differences within a row are represented by differing superscript letters.

**Table 4 foods-10-00587-t004:** Mean Just-About-Right (JAR) ratings and subsequent influence on liking ratings (penalty analysis) (WPeBS: whey permeate beverage sweetened; WPBS: whey protein beverage sweetened).

	Overall (*n* = 40)	Significance of Sample (*p* Value)	Penalty Analysis			
			Too Little		Too Much	
			Mean Drop	Frequency (%)	Mean Drop	Frequency (%)
**JAR Flavour**						
WPeBS	2.8 ± 0.1	0.82	1.48 #	25	1.21	15
WPBS	2.9 ± 0.1		1.34 #	25	2.54	15
**JAR Thickness**						
WPeBS	2.6 ± 0.1	0.17	0.11	48	−1.18 *	5
WPBS	2.8 ± 0.1		0.71	35	3.40 *	13

# represents a significant difference (*p* < 0.05) within a sample in liking compared with mean liking rating, where the sample was considered Just-About-Right; * denotes size of the group lower than 20% of population. Frequency (%) is the % of volunteers within each group (too little or too much).

**Table 5 foods-10-00587-t005:** Volunteers’ counts of whey beverage preference (WPeB: whey permeate beverage; WPeBS: whey permeate beverage sweetened; WPB: whey protein beverage; WPBS: whey protein beverage sweetened).

Pair Number	Sample	Preference	Significance of Sample (*p* Value)
1	WPeBS	24	
1	WPBS	16	0.13
2	WPB	5	
2	WPBS	35	<0.0001
3	WPeB	5	
3	WPeBS	35	<0.0001

**Table 6 foods-10-00587-t006:** Examples of volunteers’ comments relating to whey beverages (WPeBS: whey permeate beverage sweetened; WPBS: whey protein beverage sweetened).

Sample	Comments and Volunteers Details
WPeBS	“Thin, almost like drinking water (v2, female, aged 28). Nice sweet taste, but not too strong (v4, male, aged 26). There wasn’t much flavour to detect (v6, female, aged 25). Texture was OK (v9, female, aged 21)”
WPBS	“Very soothing (v1, male, aged 27). Smooth texture, bit mouthdrying (v4, male, aged 26). It is quite powdery (v7, female, aged 24). A bit too watery and thin (v30, male, aged 19)”

## Data Availability

The data presented in this study is available on request from the corresponding author. The data will be deposited in the University of Reading Research Data Archive on completion of V.N. PhD thesis.

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
