# Peer review of "Whey Protein Derived Mouthdrying Found to Relate Directly to Retention Post Consumption but Not to Induced Differences in Salivary Flow Rate"

_foods, 2021, doi:10.3390/foods10030587_

Round 1

Reviewer 1 Report

The manuscript “Whey protein derived mouthdrying found to relate directly to mucoadhesion but not to induce differences in salivary flow rate” is an interesting article, trying to look for salivary aspects that may be responsible for the mouth dry effect caused by whey protein beverages. I have some minor concerns that would like to see addressed.

One of them is the title. I think authors did not an experimental evaluation of mucoadhesion. At least not directly. Attractive forces between mucous part of the mouth and whey beverage was not measured, neither other for of quantifying mucoadhesion was used. So, I think that what authors did was to relate saliva flow rate and total protein dynamics with dry mouth perception induced by whey protein beverage. So, I suggest to change the title accordingly.

Points in the manuscript:

Material and methods – point 2.3 Model beverage preparation – the two beverages present several differences appart from containing protein, or not, namely the amout uf sugar added. Please explain better why the formulations were those. What the authors wanted to be different between beverages and what they want to be the same (as control).

Table 2 – if whey permeate beverage was sweetened with sucrose, why sugar amount is not present in the WPeBS? It can be something that I did not understand, but please clarify.

Line 223 – “Saliva samples were collected tubes” – I don’t understand this sentence. I suppose there is some mistake. Please correct.

Line 240 – “All saliva samples were weighed before and after collection”. Do you mean the “tubes” were weighed?

Results

Line 367 – “there was a tendency” – can you please specify, with p-value?

Discussion

Line 397 – “These studies were however carried out without a non-protein source” – This sentence is a little confusing. I think the authors mean that this study was carried out bout using a protein and a non-protein source. Only with this control they are able to assume that the effect is done to the protein.

Line 514 - 523 – I think the type of results obtained in this study (this experimental design) do not allow to conclude that mouthdrying is due to mucoadhesion. I suggest the authors say “Our study reinforces the idea…” or something like this. Moreover, the last sentence, suggesting that further work is needed, goes in line with my concern. Please re-phrase this paragraph because as it is it appears that the first and the last sentences are contradictory.

I miss some discussion about what may mean the increase in saliva protein concentration after ingesting the whey-protein beverage. I think the authors need also to consider that higher amount of protein in the saliva collected right after may mean that some residual protein is in saliva, and that this does not necessarily mean higher mucoadhesion. Any other proteinaceous beverage, that do not cause dry mouth may also result in higher amount of protein in the residual saliva after consumption. The authors did not control for this, so we do not know. Moreover, only the total amount of protein was evaluated, so it is not possible to know what types of proteins are present… This can be introduced as one limitation of the study.

Author Response

Thank you for your reviewing our manuscript.

We have addressed all comments and subsequent changes have been highlighted in yellow in the manuscript. Below is a summary of the changes made.

Reviewer One

The manuscript “Whey protein derived mouthdrying found to relate directly to mucoadhesion but not to induce differences in salivary flow rate” is an interesting article, trying to look for salivary aspects that may be responsible for the mouth dry effect caused by whey protein beverages. I have some minor concerns that would like to see addressed.

One of them is the title. I think authors did not an experimental evaluation of mucoadhesion. At least not directly. Attractive forces between mucous part of the mouth and whey beverage was not measured, neither other for of quantifying mucoadhesion was used. So, I think that what authors did was to relate saliva flow rate and total protein dynamics with dry mouth perception induced by whey protein beverage. So, I suggest to change the title accordingly.

Response: Title has been changed to “Whey protein derived mouthdrying found to relate directly to retention post consumption but not to induced differences in salivary flow rate”

Points in the manuscript:

Material and methods – point 2.3 Model beverage preparation – the two beverages present several differences appart from containing protein, or not, namely the amout uf sugar added. Please explain better why the formulations were those. What the authors wanted to be different between beverages and what they want to be the same (as control).

Response: Thank you for your comment. The amount of sugar added differs between the control and protein samples to allow for the highe lactose in the whey permeate, they were matched on the relative sweetness. Section 2.3 has been rewritten for clarity.

Table 2 – if whey permeate beverage was sweetened with sucrose, why sugar amount is not present in the WPeBS? It can be something that I did not understand, but please clarify.

Response: Thank you for your comment. This has been added.

Line 223 – “Saliva samples were collected tubes” – I don’t understand this sentence. I suppose there is some mistake. Please correct.

Response: Indeed, yes, this has been corrected.

Line 240 – “All saliva samples were weighed before and after collection”. Do you mean the “tubes” were weighed?

Response: Thank you for your comment. Indeed  we weighed the tubes before and after saliva collection. The text has been amended.

Results

Line 367 – “there was a tendency” – can you please specify, with p-value?

Response: Thank you for your comment. The p values are stated in preceeding sentence, but the sentence was too long to make it one sentence. We have improved the link between the two sentences to make this clearer by adding “However” at the start of the 2nd sentence.

Discussion

Line 397 – “These studies were however carried out without a non-protein source” – This sentence is a little confusing. I think the authors mean that this study was carried out bout using a protein and a non-protein source. Only with this control they are able to assume that the effect is done to the protein.

Response: Thank you for your comment. Indeed we are saying the previous studies were carried out without a non-protein control, this has been reworded..

Line 514 - 523 – I think the type of results obtained in this study (this experimental design) do not allow to conclude that mouthdrying is due to mucoadhesion. I suggest the authors say “Our study reinforces the idea…” or something like this. Moreover, the last sentence, suggesting that further work is needed, goes in line with my concern. Please re-phrase this paragraph because as it is it appears that the first and the last sentences are contradictory.

Response: Indeed we agree with your view here and have updated the paragraph accordingly.

I miss some discussion about what may mean the increase in saliva protein concentration after ingesting the whey-protein beverage. I think the authors need also to consider that higher amount of protein in the saliva collected right after may mean that some residual protein is in saliva, and that this does not necessarily mean higher mucoadhesion. Any other proteinaceous beverage, that do not cause dry mouth may also result in higher amount of protein in the residual saliva after consumption. The authors did not control for this, so we do not know. Moreover, only the total amount of protein was evaluated, so it is not possible to know what types of proteins are present… This can be introduced as one limitation of the study.

Response: Thank you for your comment. It is indeed true that we have not measured the protein type, however we do subtract the baseline to account for salivary proteins (section 2.10 and 2.11). We have not aware of studies that have related retention of protein from other proteinaceous beverages to the perception of mouthdring (or lack of), so we cannot comment on this. However, we agree we should draw our conclusions based upon retention rather then more specifically mucoadhesion, As noted above, we have modified the final paragraph of the discussion and reworded “mucoadhesion” or “oral retention” where necessary in the Abstract, Introduction, in section 4.3 as a whole and in the Conclusion.

Reviewer 2 Report

The topic of the manuscript is focusing on a current issue, whey enriched beverages are quite popular due to their increased protein content. Consumers often expect beneficial nutritional effects and simultaneously high hedonic value, which raises a challenge to product developers.

Methodology concerns

Participants’ number was very well defined on statistical basis (line 106). However, there is an ISO guidance (ISO 11136), which suggests, that no less than 60 participants should be involved in preference studies even at the pilot level. Since in the current study the authors investigated several other parameters, it is not a key issue here.

Generally, the chosen methodology is suitable for performing the proposed research objectives. From line 188, the Authors describe the 2AFC method. I agree with them concerning the suitability of that approach to detect minor differences. However, in case of discrimination tests usually we apply repetitions (more than one sample pair / assessor) in order to reduce the probability of correct guessing. It would be an interesting information to include in this section, whether participants received repetitions or not.

At line 193 we read about the rather complex module of perception and liking test. From the sensory point of view it is a very nice mosaic of different methods (9-point scale, 5-point scale, gLMS, JAR-scale). My personal concern is about the possible confusion of this matrix toward the volunteer panelists. Did they really understood the differences between the concepts of the scales? Were there any data losses due to the heterogeneity of the testing?

Author Response

Thank you for your reviewing our manuscript.

We have addressed all comments and subsequent changes have been highlighted in yellow in the manuscript. Below is a summary of the changes made.

 Reviewer Two

The topic of the manuscript is focusing on a current issue, whey enriched beverages are quite popular due to their increased protein content. Consumers often expect beneficial nutritional effects and simultaneously high hedonic value, which raises a challenge to product developers.

Methodology concerns

Participants’ number was very well defined on statistical basis (line 106). However, there is an ISO guidance (ISO 11136), which suggests, that no less than 60 participants should be involved in preference studies even at the pilot level. Since in the current study the authors investigated several other parameters, it is not a key issue here.

Response: Thank you for your comment. You are correct that our power calculations were based upon perception not preference. However the main reason to carry out a preference test was to ensure that the addition of sugar and vanilla to the drinks had increased the palatability and, as can be seen in Table 5, as these differences are large the study was sufficiently powered to detect this significant difference,

Generally, the chosen methodology is suitable for performing the proposed research objectives. From line 188, the Authors describe the 2AFC method. I agree with them concerning the suitability of that approach to detect minor differences. However, in case of discrimination tests usually we apply repetitions (more than one sample pair / assessor) in order to reduce the probability of correct guessing. It would be an interesting information to include in this section, whether participants received repetitions or not.

Response: Thank you for your comment. Indeed it is debatable whether repetitions within an individual should be used or not in discrimination testing. Where the aim is to investigate individual results (and differences between individual results) then replication is needed, however where it is the sample that is under investigation rather than the individual, the repetitions to overcome the chance probability can be through the use of single tests over multiple subjects. We have updated the manuscript to make it clear that we have used a single paired comparison test for each individual.

At line 193 we read about the rather complex module of perception and liking test. From the sensory point of view it is a very nice mosaic of different methods (9-point scale, 5-point scale, gLMS, JAR-scale). My personal concern is about the possible confusion of this matrix toward the volunteer panelists. Did they really understood the differences between the concepts of the scales? Were there any data losses due to the heterogeneity of the testing?

Response: Thank you for your comment. We carried out appropriate training and provided clear instructions to prevent data losses due the heterogeneity of testing.

This manuscript is a resubmission of an earlier submission. The following is a list of the peer review reports and author responses from that submission.